# Influenza Infection in Ferrets with SARS-CoV-2 Infection History

Caroline Vilas Boas de Melo,[a] Florence Peters,[a] Harry van Dijken,[a] Stefanie Lenz,[a]* Koen van de Ven,[a]§ Lisa Wijsman,[a] Angéla Gomersbach,[b] Tanja Schouten,[b] Puck B. van Kasteren,[a] Judith van den Brand,[c] Jørgen de Jonge[a]

[a]National Institute for Public Health and the Environment (RIVM), Bilthoven, The Netherlands
[b]Animal Research Centre, Poonawalla Science Park, Bilthoven, The Netherlands
[c]Faculty of Veterinary Medicine, University Utrecht, Utrecht, The Netherlands

**ABSTRACT** Nonpharmaceutical interventions (NPIs) to contain the SARS-CoV-2 pandemic drastically reduced human-to-human interactions, decreasing the circulation of other respiratory viruses, as well. Consequently, influenza virus circulation, which is normally responsible for 3 to 5 million hospitalizations per year globally, was significantly reduced. With the downscaling of the NPI countermeasures, there is a concern for increased influenza disease, particularly in individuals suffering from postacute effects of SARS-CoV-2 infection. To investigate this, we performed a sequential influenza H1N1 infection 4 weeks after an initial SARS-CoV-2 infection in ferrets. Upon H1N1 infection, ferrets that were previously infected with SARS-CoV-2 showed an increased tendency to develop clinical signs, compared to the control H1N1-infected animals. A histopathological analysis indicated only a slight increase for type II pneumocyte hyperplasia and bronchitis. Thus, the effects of the sequential infection appeared minor. However, ferrets were infected with B.1.351-SARS-CoV-2, the beta variant of concern, which replicated poorly in our model. The histopathology of the respiratory organs was mostly resolved 4 weeks after the SARS-CoV-2 infection, with only reminiscent histopathological features in the upper respiratory tract. Nevertheless, SARS-CoV-2 specific cellular and humoral responses were observed, confirming an established infection. On account of a modest trend toward the enhancement of the influenza disease, even upon a mild SARS-CoV-2 infection, our findings suggest that a stronger SARS-CoV-2 infection and its consequent, long-term effects could have a greater impact on the outcome of disease after a sequential influenza infection. Hence, the influenza vaccination of individuals suffering from postacute SARS-CoV-2 infection effects may be considered an avertible measure for such a scenario.

**IMPORTANCE** During the COVID-19 pandemic, the use of face masks, social distancing, and isolation were effective not only in decreasing the circulation of SARS-CoV-2 but also in reducing other respiratory viruses, such as influenza. With fewer restrictions currently in place, influenza is slowly returning. In the meantime, people who are still suffering from long-COVID could be more vulnerable to an influenza virus infection and could develop a more severe influenza disease. This study provides directions to the effect of a previous SARS-CoV-2 exposure on influenza disease severity in a ferret model. This model is highly valuable to test sequential infections under controlled settings for translation to humans. We could not induce clear long-term COVID-19 effects, as the SARS-CoV-2 infections in the ferrets were mild. However, we still observed a slight increase in influenza disease severity compared to ferrets that had not encountered SARS-CoV-2 before. Therefore, it may be advisable to include long-COVID patients as a risk group for influenza vaccination.

**KEYWORDS** COVID-19, SARS-CoV-2, VOC, influenza, ferret model, sequential infections

Address correspondence to Jørgen de Jonge, jorgen.de.jonge@rivm.nl.

*Present address: Stefanie Lenz, MSD, Boxmeer, The Netherlands.

§Present address: Koen van de Ven, DICA (Dutch Institute for Clinical Auditing), Leiden, The Netherlands.

The authors declare no conflict of interest.

2 years after the emergence of SARS-CoV-2 in late 2019, the causative agent of COVID-19 is still an important global health and economic problem. In order to mitigate virus transmission, a set of nonpharmaceutical interventions (NPIs), such as

the use of face masks, physical distancing, and travel restrictions, were adopted worldwide, leading to a considerable decrease in social interactions (1). The unprecedented reduced human interaction was not only effective against SARS-CoV-2 transmission but also resulted in an almost complete absence of the circulation of several other respiratory viruses (2–4). For influenza virus, for example, the recorded peak of over 40,000 weekly cases in the global 2019 to 2020 influenza season dramatically decreased in incidence from March 2020 to essentially zero in the following months, leading to the absence of the global 2020 to 2021 influenza season (5). With the relaxation of NPIs and the subsequent renewed circulation of respiratory viruses, there is a concern that influenza infection might lead to more severe disease in individuals previously exposed to SARS-CoV-2. There are at least two reasons for this: (i) the interrupted seasonal exposure has resulted in a reduction of immunity in the population, which is therefore more susceptible (6), and (ii) the residual effects of COVID-19 can have a negative clinical impact on a subsequent respiratory viral infection, such as influenza.

Fatigue, dyspnea, and chest pain are commonly reported after a cleared SARS-CoV-2 infection (7). These long-term effects of COVID-19, called postacute COVID-19 or long-COVID syndrome, were associated with an immune signature of epithelial injury and tissue repair in the airways (8) and are possible clinical repercussions of the pulmonary sequelae of a SARS-CoV-2 infection (9, 10). The only evidence for increased disease in a situation where SARS-CoV-2 and influenza cocirculated goes back to the beginning of the SARS-CoV-2 pandemic, when the influenza virus was still circulating. It was then observed that a higher risk for a severe outcome was associated with influenza coinfection in lethal cases of COVID-19 (11–14). In the current stage, however, it is still unclear whether a recent SARS-CoV-2 infection history could negatively impact a subsequent influenza infection.

Animal models are an ideal tool to investigate combinations of infections under controlled conditions. Multiple animal models of COVID-19 are currently available (15), but not all of them are equally suitable for modeling influenza disease. So far, coinfection studies with SARS-CoV-2 and influenza A virus (FLUAV) in hamster and K18-ACE2 mouse models indicate more severe pneumonia (16–18) and confirm the initial findings in humans (19). Ferrets coinfected with SARS-CoV-2 and FLUAV developed increased weight loss and enhanced inflammation in the nasal cavity and lungs (20). In transgenic hACE2 mice, upon a sequential infection with H1N1 with a brief time interval (7 dpi) during the convalescent SARS-CoV-2 infection phase (14 dpi) (21) more severe lung damage was observed. Little is known, however, about the effects of an influenza infection following a resolved SARS-CoV-2 infection or during postacute COVID-19, which is the more likely scenario.

Ferrets are the best small animal model for influenza infection and disease, and they are also susceptible to SARS-CoV-2 infection, albeit reproducing only mild or nonclinical COVID-19 (15, 22, 23). Nevertheless, follicular hyperplasia in the higher airways of the lung and inflammatory infiltration in the nasal cavity have been observed 21 days after experimental infection with SARS-CoV-2, after the viral infection had long been cleared (24). These observations could reflect, in part, the long-term disease (postacute COVID-19) characterized by prolonged respiratory complaints and fatigue. We experimentally infected ferrets with the SARS-CoV-2 beta variant (B.1.351), a variant of concern (VOC) at the time these experiments were performed. The ferrets were followed for up to 4 weeks postinfection, with the goal being to resemble postacute COVID-19, and they were then infected with an H1N1 influenza virus. Our aim was to investigate the impact of the previous exposure to SARS-CoV-2 on the severity of disease and on the pathology of respiratory organs upon a sequential influenza (H1N1) infection.

## RESULTS

**SARS-CoV-2 infection in ferrets.** To evaluate the impact of a previous SARS-CoV-2 infection on a sequential influenza infection during the postacute phase, we first studied viral replication and clinical signs in two groups of male ferrets that were infected

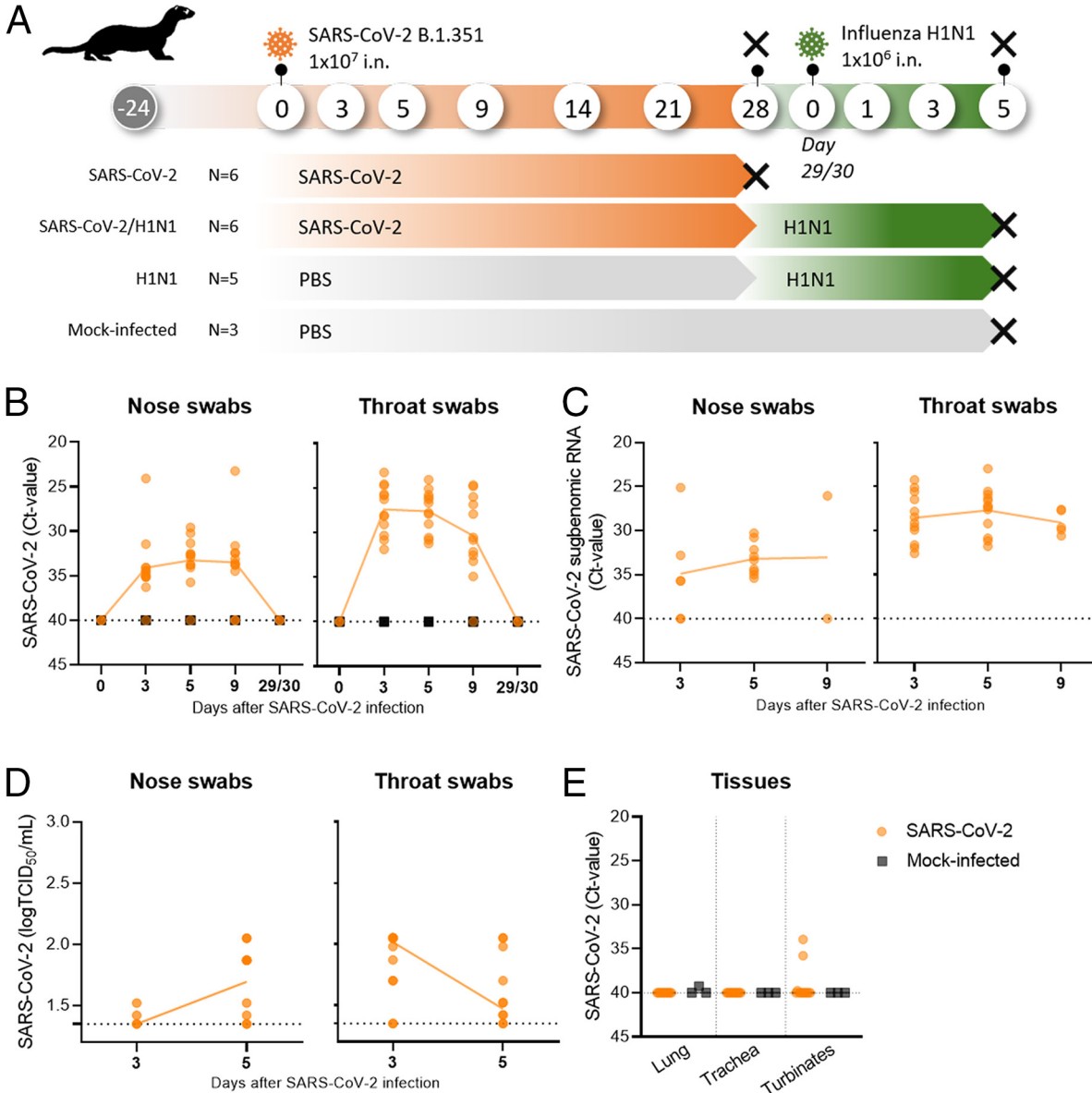

FIG 1 Study design and SARS-CoV-2 (B.1.351, beta VOC) viral load. (A) Male ferrets were infected intranasally (i.n) with SARS-CoV-2 (B.1.351, beta VOC), H1N1, or SARS-CoV-2 and H1N1 in sequence. Control groups were administered i.n. with phosphate-buffered saline (PBS, mock-infected). Two groups of six ferrets each were infected with beta VOC on day 0. One of these groups was followed for up to 28 days postinfection (dpi) and then euthanized (black cross). The other group was sequentially infected with H1N1 on days 29/30. One group of five ferrets was single-infected with H1N1 at the same time point. The H1N1-infected and mock-infected ferrets were followed for up to 5 days after the H1N1 infection and then euthanized. (B–E) Viral load and virus replication of all beta VOC-infected ferrets (n = 12, orange dots) and mock-infected ferrets (n = 3, black squares). (B) Beta VOC viral RNA detected at 0, 3, 5, 9, and 29/30 dpi. Only the remaining six beta VOC-infected ferrets were tested to ensure the absence of the virus at the moment of receipt of the H1N1 infection (day 29/30). Beta VOC subgenomic RNA detected via RT-PCR (C) and viral replication detected via TCID$_{50}$ assay (D) in nose and throat swabs. (E) Beta VOC viral RNA in lung, trachea, and nasal turbinates at 28 dpi via RT-PCR. Horizontal dotted lines depict the limit of detection via RT-qPCR, set at a Ct-value of 40 (B, C, and E) and 1.3 log TCID$_{50}$/mL for the TCID$_{50}$ assay. Connecting lines depict the mean, and dots represent individual observations.

with the SARS-CoV-2 beta VOC during a 4 week period. In addition, we analyzed the humoral and cellular immune responses and sacrificed one of the two groups at day 28 to determine the pathological status of the respiratory tract at the day prior to the sequential influenza infection. The study was then continued with an influenza infection of naive ferrets and the remaining group of ferrets that were infected with SARS-CoV-2 beta 4 weeks earlier (Fig. 1A). A control group consisted of mock-infected ferrets.

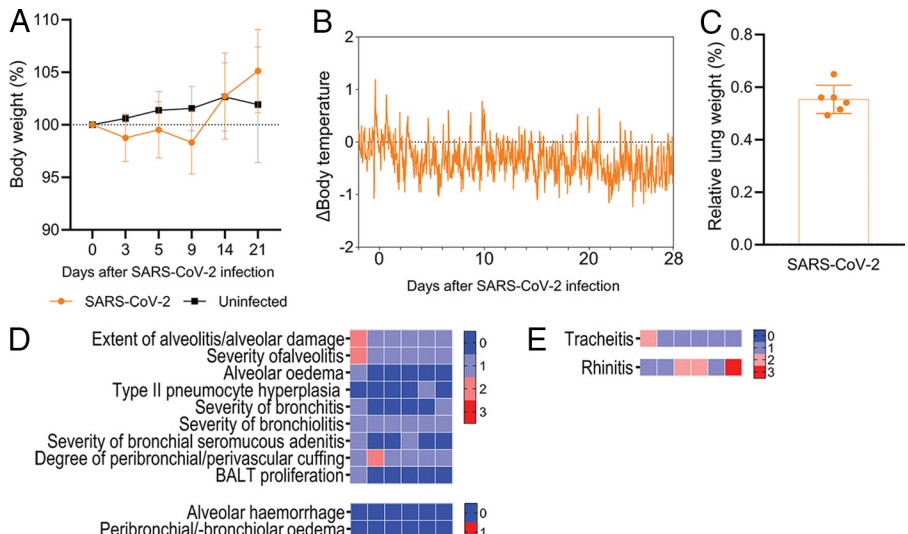

**FIG 2** Clinical signs and histopathology in the postacute SARS-CoV-2 (B.1.351, beta VOC) experimental model. (A) Percentual body weight variation relative to baseline values (day 0) of beta VOC-infected ferrets ($n = 12$, orange dots) and mock-infected ferrets ($n = 3$, gray squares) on 0, 3, 5, 9, 14, and 21 days postinfection (dpi). (B) Differences (Δ) in body temperature up to 28 dpi from the beta VOC-infected ferrets ($n = 5$) relative to the baseline measurements recorded over a period of 5 days prior to infection. (C) Relative lung weight (RLW, %) of the body weight upon euthanasia on 28 dpi of the beta VOC-infected ferrets ($n = 6$). The RLW is comparable to RLW of mock-infected ferrets (Fig. 4D). (D and E) Categorical heatmaps of histopathology represented in semiquantitative intensity scores, with color shading from blue (0) to red (3), or in qualitative scores of absence (0, blue) or presence (1, red), where indicated. (D) Lung histopathology and (E) histopathology scoring of the inflammation of the trachea (tracheitis) and nasal turbinates (rhinitis) of six SARS-CoV-2-infected ferrets. Data are visualized as the mean with the standard deviation, and dots represent individual observations. For reference, the data for the body temperature, gross pathology, and histopathology of the mock-infected ferrets can be found in Fig. 4 and 5.

SARS-CoV-2 viral RNA was detected in nose and throat swabs of all 12 infected ferrets between 3 dpi and 9 dpi, with the exception of 1 ferret that was negative at 3 dpi and another ferret that was negative at 5 dpi in a nose swab (Fig. 1B). At 9 dpi, SARS-CoV-2 viral RNA was not detected in the nose swabs of three ferrets and in a throat swab of one ferret. Viral RNA was not detected in the mock-infected ferrets and was no longer present in any of the animals at 29 dpi. The detection of subgenomic viral RNA evidenced a viral infection in the nasal cells at 3 dpi (3 out of 12 ferrets) and 5 dpi (9 out of 12 ferrets) (Fig. 1C). Furthermore, all of the ferrets were positive for subgenomic RNA in throat swabs at both 3 dpi and 5 dpi.

Nevertheless, a productive infection was barely established, since low or sometimes absent viral titers were observed via a 50% tissue culture infectious dose ($TCID_{50}$) determination in nose swabs (peak at day 5, $1.6 \pm 0.2$ log $TCID_{50}$/mL) and throat swabs (peak at day 3, $1.8 \pm 0.2$ log $TCID_{50}$/mL) (Fig. 1D). To confirm that SARS-CoV-2 had disappeared prior to the influenza infection, we determined the viral load in the respiratory tissues. As expected, viral RNA was not detected in the trachea or the lung at day 28, but negligible low copy numbers were found in the nasal turbinates of two out of six ferrets (Fig. 1E).

The clinical signs observed in the SARS-CoV-2-infected ferrets were mild and of low frequency. Between 1 and 5 days after the SARS-CoV-2 infection, increased breathing rhythm and/or decreased activity were observed in 6 out of 12 ferrets. Except for one ferret that showed mild decreased activity at 9 dpi and had not shown any other clinical signs before, all clinical signs were absent at 6 dpi (data not shown). SARS-CoV-2 infected ferrets did not present notable body weight loss in comparison to the mock-infected ferrets (Fig. 2A), nor did they develop fever, as shown by normal daily variations in body temperature (Fig. 2B). Upon euthanasia, the ferrets did not show signs of

COVID-19 in gross lung pathology, and the relative lung weight, an indication of the presence of edema secondary to inflammation in the lungs, was not increased (Fig. 2C). Rare histopathology findings in the lungs (Fig. 2D), together with moderate rhinitis in the nasal turbinates (Fig. 2E), reflect an absent to slight postacute pathology of the respiratory organs 28 days after the SARS-CoV-2 infection.

**Immune response in SARS-CoV-2 infection in ferrets.** Despite the low level of replication of the SARS-CoV-2 beta VOC (Fig. 1B–E), the ferrets did develop cellular immune responses, as shown by the restimulation of peripheral blood mononuclear cells (PBMCs) with SARS-CoV-2 peptide pools and live virus in IFN-$\gamma$-ELISpot (Fig. 3A). IFN-$\gamma$ production was observed in the SARS-CoV-2 beta VOC-infected ferrets against both live alpha and beta VOCs, reflecting the similarity between these variants. This was sustained by the finding of IFN-$\gamma$ responses upon stimulation with peptide pools of spike protein (S) from the original SARS-CoV-2 strain. Additionally, we found an IFN-$\gamma$ response upon stimulation with nucleocapsid protein (N), but not with nonstructural protein 12 (nsp12), which contains the RNA-dependent RNA polymerase (RdRp). No responses were detected in the mock-infected ferrets, or in the PBMCs stimulated with negative controls (mumps virus and the HIV peptide pool). In addition to the ELISpot assay, we investigated the phenotype of the IFN-$\gamma$-producing T cells via flow cytometry at 28 dpi (Fig. 3B and C). High IFN-$\gamma$ responses in the PBMCs against SARS-CoV-2 beta VOC were observed in both CD4 and CD8 T cells after live virus stimulation. No responses were detected after the stimulation with peptide pools spanning for the S protein (amino acid ranges 1 to 158 and 159 to 315) in the CD8 T-cells, as the response was not distinguishable from the background (medium) and the negative-control (HIV peptide pool). For the CD4 T cells, the IFN-$\gamma$ response was distinctively found against the second portion (159 to 315) of the S protein. The antibody responses against the S and RBD of the SARS-CoV-2 beta VOC were found from 14 dpi and reached a plateau on 21 to 28 dpi (Fig. 3D). Taken together, the viral replication and immune data indicate that a reproductive but low replication was established in the majority of the ferrets, as only exposure to virus does not lead to an immune response (24).

**Influenza H1N1 infection and disease in ferrets with previous SARS-CoV-2 exposure.** We next investigated whether influenza virus-induced pathology and disease are enhanced during the postacute phase of SARS-CoV-2 (SARS-CoV-2/H1N1 group) in comparison to H1N1 influenza virus infection alone (H1N1 group) (Fig. 1A). The H1N1 viral load and replication were assessed on 0, 1, 3, and 5 dpi with H1N1 influenza virus via TCID$_{50}$ and real-time reverse-transcription polymerase chain reaction (RT-qPCR) (Fig. 4A and B). Viral RNA and virus replication were observed as early as 1 dpi, in both nose and throat swabs. Upon euthanasia on 5 dpi, H1N1 RNA was found in the lung, trachea and nasal turbinates, while viable virus was only recovered from the nasal turbinates via the TCID$_{50}$ assay (Fig. 4C). Overall, the viral load and viral replication were similar between groups, irrespective of prior SARS-CoV-2 infection, and they were absent in the mock-infected ferrets. Gross pathology showed signs of inflammation in the lungs of all animals infected with H1N1, and this was also absent in the mock-infected ferrets (data not shown). The relative lung weight, representing the edematous lung, was increased in all of the H1N1-infected ferrets compared to the mock-infected ferrets, irrespective of previous SARS-CoV-2 infection (Fig. 4D). All of the H1N1-infected ferrets lost 6 to 8% of their body weight from day 3 of H1N1 infection, which was not significantly different between groups (Fig. 4E). Increased body temperature in the first 3 days of H1N1 infection was observed in a similar curve between the groups (Fig. 4E). Taken together, the viral burden and gross pathology did not indicate that influenza disease severity increased upon prior exposure to SARS-CoV-2. Nonetheless, clinical signs that are characteristic of influenza disease were more persistent in ferrets that received an H1N1 influenza virus infection consecutive to a SARS-CoV-2 infection (Fig. 4G). In the 5 days following the H1N1 infection, decreased activity and labored breathing were detected from day 4, whereas nasal discharge and sneezing were not detected. The relative frequency of decreased activity was similar in both groups of H1N1-infected ferrets on day 4. Labored breathing was observed in all of the

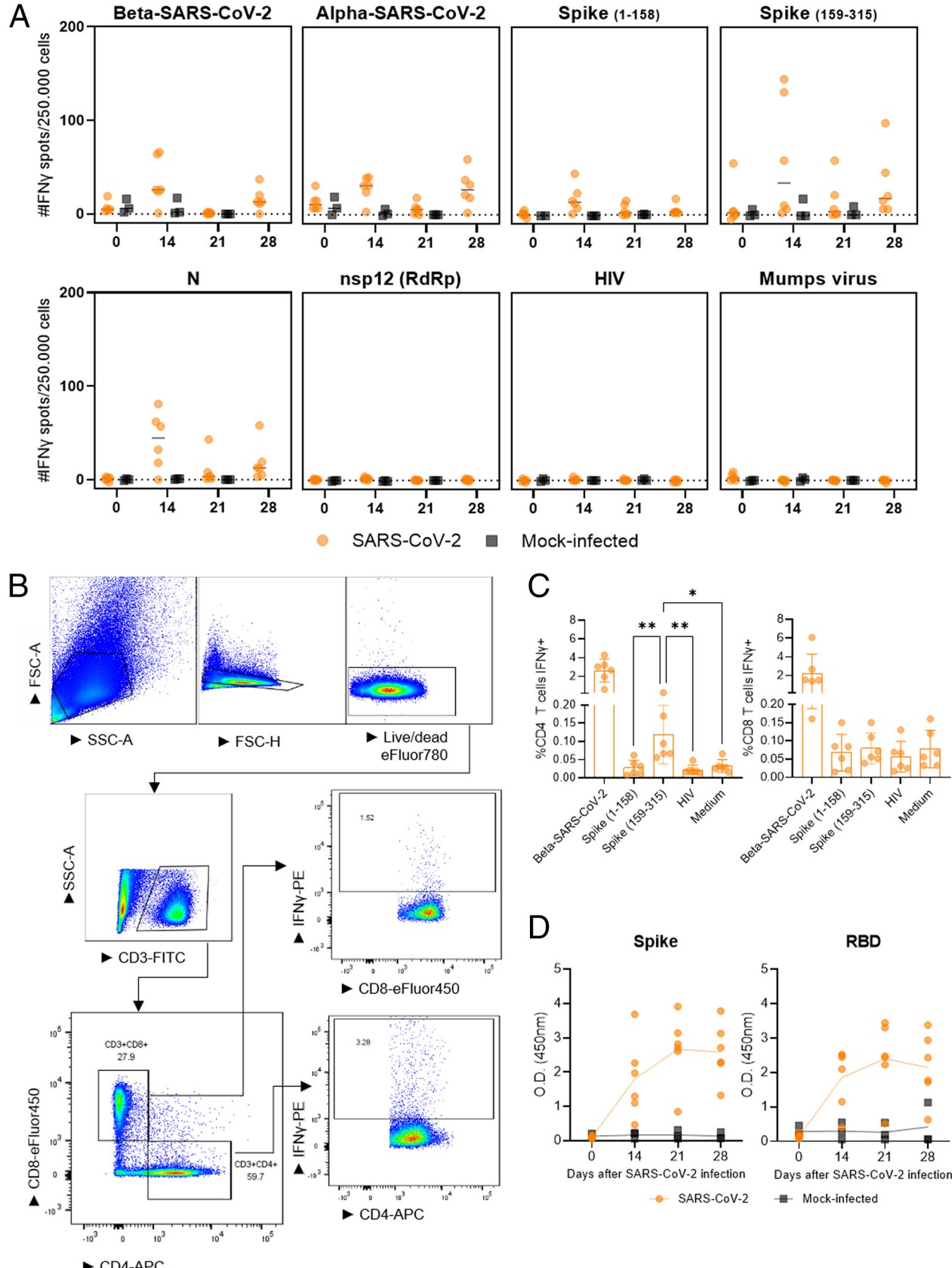

**FIG 3** Cellular and humoral responses in the SARS-CoV-2 (B.1.351, beta VOC) experimental model. (A) Cellular responses in PBMC on 0, 14, 21, and 28 days postinfection (dpi) measured by IFN-γ ELISpot against the SARS-CoV-2 alpha VOC (with a culture-acquired mutation in the ORF1a

H1N1-infected ferrets with previous SARS-CoV-2 exposure (six out of six ferrets, 100%) in comparison to four out of five ferrets (80%) ferrets exposed to H1N1 only on day 4. On day 5, clinical signs were only observed in the sequentially infected ferrets (SARS-CoV-2/H1N1), with more pronounced labored breathing (five out of six ferrets, 83%) and decreased behavioral activity (two out of six ferrets, 33%), whereas the H1N1-infected ferrets without SARS-CoV-2 prior exposure were free of clinical signs.

**Pathology of H1N1 infection and SARS-CoV-2 in respiratory organs of ferrets.** Although we did not observe any strong pathological effects in the lungs on 1 day (D28) prior to influenza infection in ferrets with a SARS-CoV-2 infection history (Fig. 2D and E), a previous SARS-CoV-2 infection did appear to marginally worsen influenza disease signs. Therefore, we wondered whether differential pathology could still be observed in the respiratory tract between naive and SARS-CoV-2-exposed ferrets. In general, moderate to severe broncho-interstitial pneumonia, bronchoadenitis, a thickening of the alveolar septa, and large amounts of edema in the alveolar lumina, which are characteristic of influenza disease, were observed (Fig. 5A). Regardless of prior SARS-CoV-2 exposure, the histopathology findings were homogenous between the groups, except for a trend of increased type II pneumocyte hyperplasia, especially in the cranial lobe of the lungs (Fig. 5A and B), and moderate bronchitis found more prominently in the caudal lobe in the SARS-CoV-2 group that was sequentially infected with H1N1 (Fig. 5C). Severe rhinitis was observed in all of the H1N1-infected ferrets (Fig. 5D), and it was also present in moderate intensity in the SARS-CoV-2-infected ferrets, long after the SARS-CoV-2 infection was cleared (Fig. 2E). Mild to moderate tracheitis was also observed (Fig. 5D). Taken together, these results suggest that a previous mild SARS-CoV-2 beta VOC infection does not notably enhance the respiratory tract pathology induced by a sequential infection with influenza.

## DISCUSSION

In this study, we investigated whether a previous exposure to SARS-CoV-2 enhances influenza disease during the postacute phase of SARS-CoV-2 infection. Following a resolved, low-level infection with the beta VOC of SARS-CoV-2, ferrets showed a tendency of increased influenza clinical signs and minor effects in histopathology upon a sequential H1N1 infection, compared to H1N1 infection without prior exposure to SARS-CoV-2. It remains to be determined whether a robust SARS-CoV-2 infection would have a greater impact on a sequential H1N1 infection.

It has been shown that coinfection between influenza and SARS-CoV-2 results in a more severe disease in animal models through enhanced lung damage (17, 18, 20, 25). This simultaneous infection was also reported in human patients (11, 12, 26). In our study, we looked at the effects of a sequential influenza infection during the postacute phase of SARS-CoV-2 (28 days), as an influenza infection is less likely to happen simultaneously or shortly after a SARS-CoV-2 infection, compared to an infection during the span of time in which humans suffer from long-COVID. In the current study, we established a reproductive H1N1 influenza virus infection with clinical signs, viral replication kinetics, and histopathology characteristic of an infection with this subtype (27). The intensity and number of histopathological observations in the respiratory tract, however, was not clearly different between ferrets that were previously infected with SARS-CoV-2 and ferrets that were not. Nevertheless, a trend for more severe bronchitis and type II

**FIG 3** Legend (Continued)
region), the beta VOC, and SARS-CoV-2 peptide pools spanning the spike protein (amino acid ranges 1 to 158 and 159 to 315), nucleocapsid protein (N), and RNA-dependent RNA polymerase (RdRp) nonstructural protein 12 (nsp12). Stimulations with mumps virus and the HIV peptide pool were used as negative controls for the virus and peptide pool stimulations, respectively. (B) Gating strategy of immunophenotyping of IFN-$\gamma$-producing CD3$^+$CD4$^+$ and CD3$^+$CD8$^+$ T cells via flow cytometry and (C) the relative (%) quantification of IFN-$\gamma$-producing T cells on 28 dpi. The mock-infected group was not sampled due to its continuation in the study, as it served as the control group for the subsequent H1N1 infection. (D) Humoral responses were measured via enzyme-linked immunosorbent assay (ELISA). The optical density (OD) values evidence antibody responses against the spike protein and receptor binding domain of the beta VOC on 0, 14, 21, and 28 dpi. The data are visualized as the median (horizontal line, panel A) or the mean (bar or connecting line, panels C and D) with the standard deviation, and dots represent individual observations. *, $P = 0.01$; **, $P < 0.007$; calculated from analyses of variance (ANOVAs) and Tukey's multiple-comparison tests.

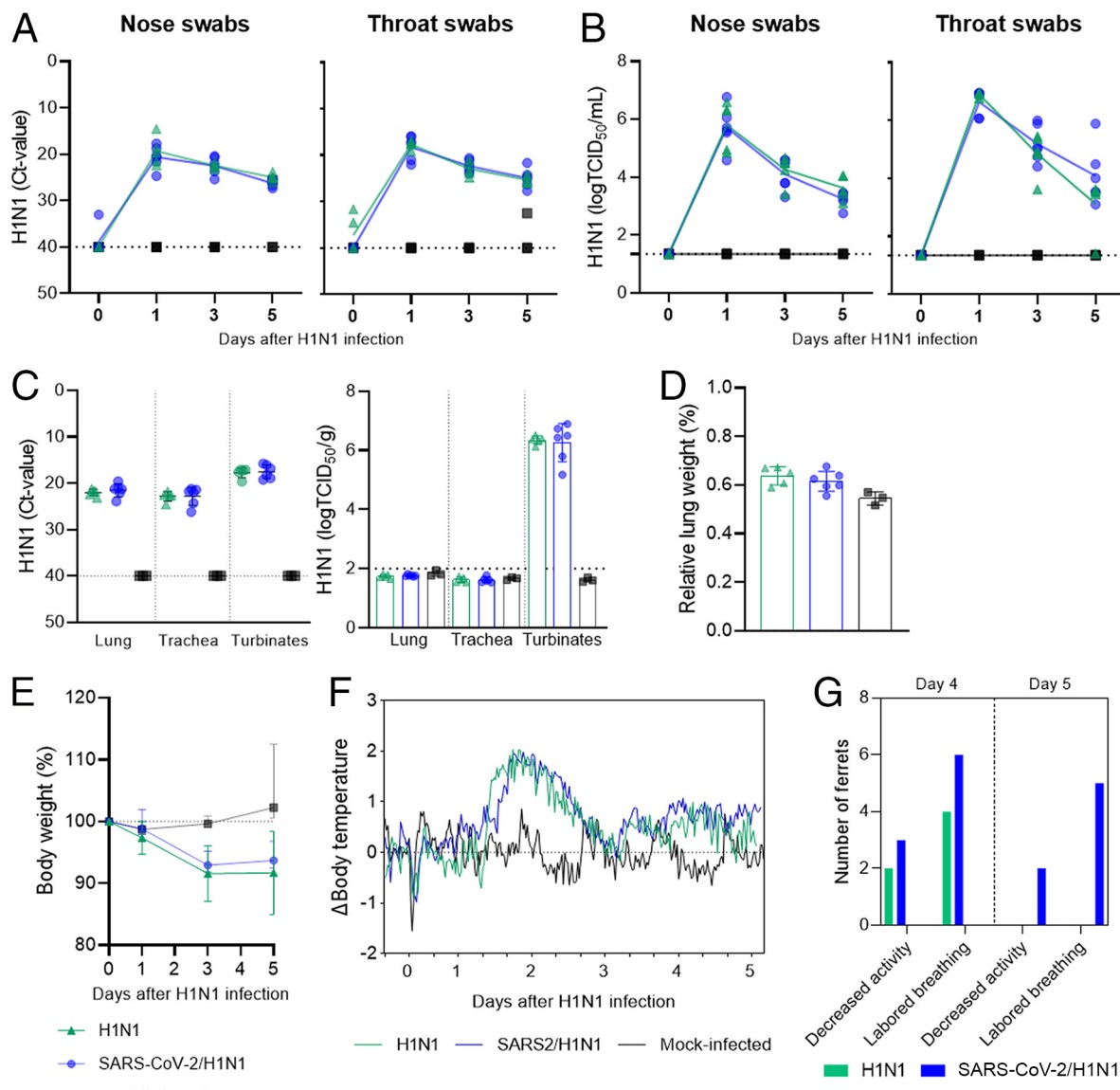

**FIG 4** H1N1 influenza virus infection and disease in ferrets previously exposed to SARS-CoV-2 (B.1.351, beta VOC). H1N1 influenza viral load, virus replication, and clinical disease in ferrets infected with the H1N1 influenza virus only (H1N1, $n = 5$, green triangles), in ferrets previously infected with the SARS-CoV-2 beta VOC (SARS-CoV-2/H1N1, $n = 6$, blue dots), and in mock-infected ferrets ($n = 3$, black squares). (A) H1N1 viral RNA detected via RT-qPCR and (B) H1N1 virus replication detected via $TCID_{50}$ assay in nose and throat swabs on 0, 1, 3, and 5 days postinfection (dpi). (C) H1N1 viral RNA was detected via RT-PCR, and viral replication was assessed via $TCID_{50}$ in the lung, trachea, and nasal turbinates on 5 dpi. (A–C) Horizontal dotted lines depict the limit of detection via RT-PCR and $TCID_{50}$ assay. (D) Relative lung weight (%) of the body weight upon euthanasia on 5 dpi. (E) Percentual body weight variation relative to weight on the day of infection (day 0) of H1N1-infected or mock-infected ferrets on 0, 1, 3, and 5 dpi (F). Differences (Δ) in body temperature up to 5 dpi, relative to the baseline measurements recorded 5 days prior to infection. (G) Recorded clinical signs of influenza disease on 4 and 5 dpi. Indicated are the numbers of ferrets that had decreased activity (score 1) and labored breathing (score 1), according to the scoring system described in the Materials and Methods section. The data are visualized as the mean, and the data points represent individual observations (panels A–D). Bars are set at the mean, and the standard deviation is shown (panels D and E). Significant statistical differences were not observed between test groups (H1N1 and SARS-CoV-2/H1N1) and were not compared to the control group (mock-infected).

pneumocyte hyperplasia scores was observed in ferrets that were sequentially infected with SARS-CoV-2 and then the H1N1 influenza virus, although the difference was not statistically significant. This observation could explain the tendency of more persistent alterations in clinically observed respiratory function in the sequentially infected ferrets. Both the SARS-CoV-2 and influenza viruses infect type II pneumocytes (28, 29), which could account for the hyperplasia in response to the infection in both occasions,

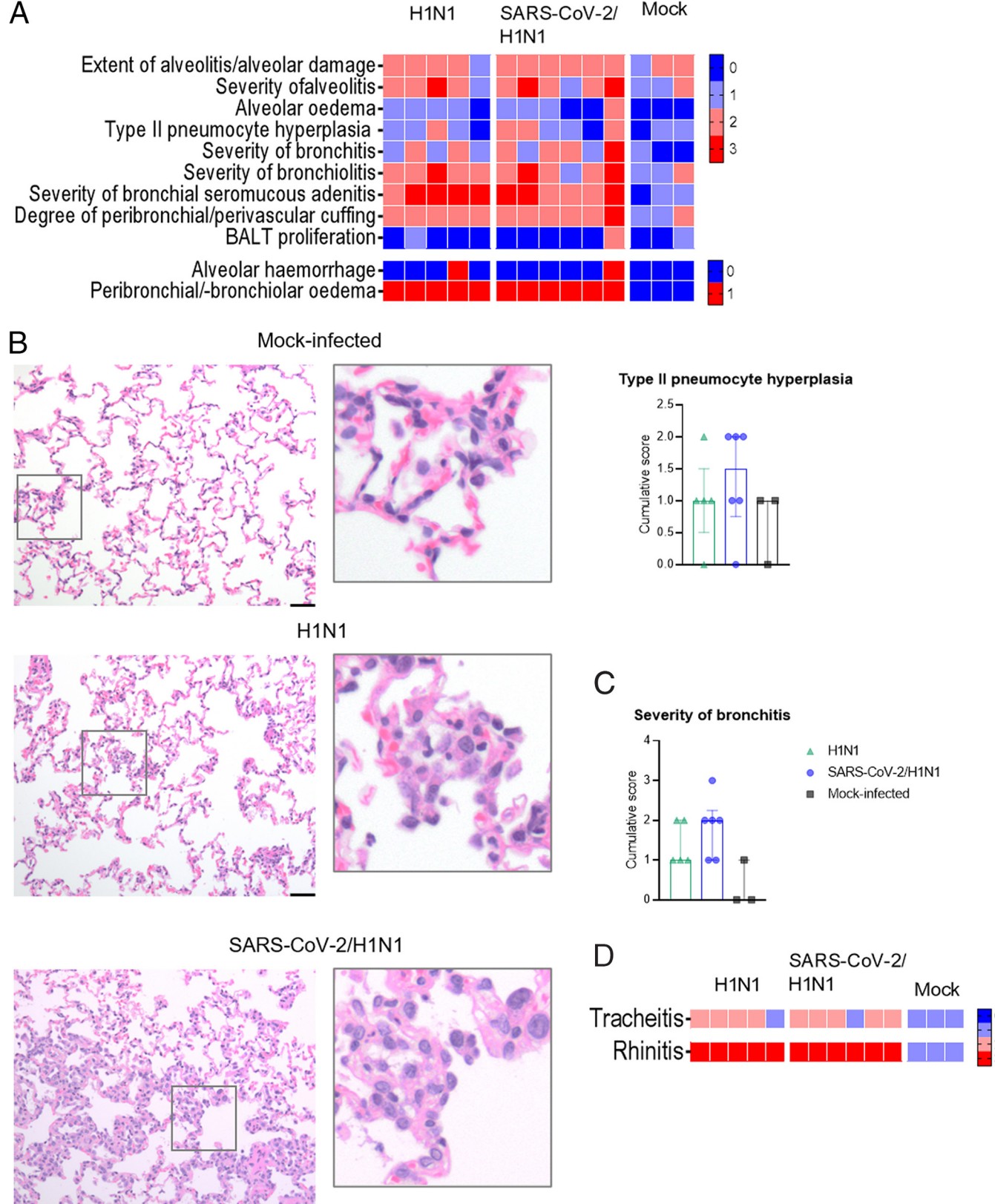

**FIG 5** Histopathology of H1N1 infection in ferrets previously exposed to SARS-CoV-2 (B.1.351, beta VOC). Categorical heatmaps of histopathology represented in semiquantitative intensity scores, with color shading from blue (0) to red (3), or in qualitative scores of absence (0, blue) or presence (1, red), where indicated. (A) Lung histopathology scoring. (B) Illustrative histopathological images of type II pneumocyte hyperplasia and magnified insets, with semiquantification represented in a cumulative score. (C) Semiquantitative cumulative score of the severity of bronchitis in ferrets infected with the H1N1 influenza virus only (H1N1, $n = 5$, green

as well as an indication that acute lung injury caused by influenza could be worsened by a previous SARS-CoV-2 infection.

Prolonged clinical symptoms of COVID-19 are considered a syndrome after a long-resolved SARS-CoV-2 infection in humans and are known as postacute COVID-19 or long-COVID (9, 10). The knowledge of the lung pathology in postacute COVID-19 cases is scarce, as most information on human lung pathology is known from acute fatal cases (30). In a previous study using SARS-CoV-2 with the D614G mutation, we observed a continued inflammatory infiltration in the upper respiratory tract and the hyperplasia of bronchus-associated lymphoid tissue in the lungs of infected male ferrets on 21 dpi (24). These late histopathology findings may reflect, in part, the extended signs and symptoms observed in human long-COVID cases. Studies performing intranasal infection with the original strain of SARS-CoV-2 showed pathology alterations in the respiratory tract on 21 dpi, after the virus was cleared, suggesting an extended effect of the infection (24, 31). Also, at 21 dpi, female ferrets were shown to develop mild (peri)bronchiole and interalveolar septal infiltration (31). In the current study, we looked at these effects on 28 dpi, as we expected that the histopathology found in other studies at 21 dpi would likely not yet be resolved. Yet, only a moderate inflammation in the nasal turbinates was observed, and no histopathology alterations were found in the lower respiratory tract. Thus, using the beta VOC, we could not induce any histopathological effects that would be indicative of long-COVID. We assume, however, that the absence of pathology in the present study is due to the low level of productive infection when using the beta VOC, since effective SARS-CoV-2 viral replication frequently resulted in pathology manifestation in ferrets in other studies (20, 23, 24, 31–33). On the other hand, the low level of virus replication and the absence of disease in our beta VOC model may represent a large part of the human population who undergo mild or asymptomatic SARS-CoV-2 infections and who are also subject to subsequent influenza infections.

Ferrets are a suitable model of COVID-19 for features such as viral replication kinetics, virus dissemination, and transmission, compared to other small animal models (31, 34). Clinical presentation is absent or mild, and histopathology findings usually show variable grades of inflammatory infiltration in the upper and/or lower respiratory tract, which is not homogeneous among reports (23, 32, 35). However, a large share of these studies relate to the original strains of SARS-CoV-2, and little is currently known about the susceptibility of the ferret to VOCs. At the time this study was conducted, the beta SARS-CoV-2 variant was one of the variants of concern and was increasing in circulation worldwide. Therefore, the beta VOC was chosen for its epidemiological relevance. One investigation reported successful replication with the beta VOC in female ferrets, with a 100-times lower dose than that which we used in this study (36). Using a high infectious dose ($10^7$ TCID$_{50}$/mL) in male ferrets, we found that the beta VOC replicated only to low levels, as measured via TCID$_{50}$. Reproductive infection was further confirmed by the detection of subgenomic viral RNA and by the clear development of a humoral and cellular immune response, which does not occur from only exposure to a virus (24). While it has been shown that the SARS-CoV-2 alpha VOC successfully replicates in ferrets (37, 38), the absence of a productive infection with the beta VOC in ferrets was recently reported by others (38). In another study, the beta VOC was outcompeted by coinfection with the alpha VOC in ferrets and thus proved to be less fit in ferrets (37). The emergence of the beta VOC is believed to have resulted from immune escape in a partly immunized population, whereas in a naive population, it proved to be less fit than the alpha variant (37, 39), which may reflect the low replication found in the naive ferrets in this study.

Regardless of the low level of productive infection of beta VOC, cellular and humoral responses were induced. IFN-$\gamma$ production was found in PBMCs upon stimulation with

**FIG 5** Legend (Continued)

triangles) or in ferrets previously infected with SARS-CoV-2 (SARS-CoV-2/H1N1, $n = 6$, blue dots) and mock-infected ferrets ($n = 3$, black squares). Bar = 50$\mu$m, H & E, 200× magnification. (D) Histopathology scoring of the inflammation of the trachea (tracheitis) and the nasal turbinates (rhinitis). The data are presented as the median and the interquartile range. Statistically significant differences were not observed between the test groups (H1N1 and SARS-CoV-2/H1N1), and comparisons were not made to the control group (mock-infected).

other SARS-CoV-2 strains, indicating that the mutations that appeared in the VOCs did not affect the cellular response. In fact, using peptide stimulations based on the original strain, N induced IFN-$\gamma$ production, but a more prominent response was found against the S protein. In line with our results, immune responses against S and N proteins are described for both ferrets and humans (24, 40, 41); however, we did not detect a response to nsp12 (RdRp). Likewise, in humans, responses to nsp12 are either absent or low (42). Taken together, these results underline our previous finding that the ferret model reflects cellular immune responses in humans at the protein level (24, 43).

The innate system has its own memory-like capability, called trained immunity. An encounter with a pathogen keeps the innate immune system in a more alert state for a period of time. Trained immunity is therefore also considered to play a role in the protection against subsequent infections by other pathogens (43). Along this line, a previous SARS-CoV-2 infection may even be beneficial for the outcome of a secondary influenza infection. In the reverse situation, protection against COVID-19 by an influenza vaccination illustrates the theory that the components of innate immunity are able to confer cross-protective immune responses (44). However, our findings of slightly enhanced pathology and clinical signs of influenza disease after a resolved SARS-CoV-2 infection indicate otherwise in this study.

In conclusion, our data suggest that a preceding SARS-CoV-2 infection may play a role in the severity of disease after a sequential influenza infection. Because a slight enhancement of influenza disease was observed upon even a mild preceding SARS-CoV-2 infection, further studies are necessary to explore the impact of a previous, more virulent SARS-CoV-2 infection and the development to long-COVID. Currently, patients suffering from long-COVID are not specifically considered to be a risk group for influenza disease. In light of the results found herein, despite being subtle, it may be advisable to preemptively include this group in the influenza vaccination campaign.

## MATERIALS AND METHODS

**Ethics statement.** All animal procedures were conducted in accordance with the European regulations for animal experimentation. The study proposal was evaluated and approved by the Animal Welfare Body of Poonawalla Science Park – Animal Research Center (Bilthoven, The Netherlands) under permit number AVD32600 2018 4765 of the Dutch Central Committee for Animal Experiments.

**Ferrets and housing.** 20 outbred male ferrets (*Mustela putorius furo*), aged 8 months, were obtained from the colony of Euroferret (Denmark). The ferrets were investigated for previous Aleutian disease, canine herpesvirus, and ferret coronavirus infections via the measurement of their antibody titers in serum via enzyme-linked immunosorbent assay (ELISA), considering a titer cutoff of an optical density (OD) value of <100 (Table S1). All of the ferrets tested negative for antibody responses measured against the SARS-CoV-2 spike protein (S), its receptor binding domain (RBD), and the nuclear protein of influenza virus via ELISA. The ferrets were semirandomly allocated to three matching groups by weight, with an average of 1.3 kg $\pm$ 0.1 kg, and were subsequently acclimated for 24 days in open cages. Prior to infection, the ferrets were moved to BSL3 isolators in groups of 3 until termination.

**Cell culture and virus isolates.** The SARS-CoV-2 beta variant (B.1.351) was isolated from a COVID-19 patient (hCoV-19/Netherlands/NH-RIVM-20159/2021, nextclade 20H/501Y.V2). Influenza A virus (H1N1) A/Michigan/45/2015 was obtained from the National Institute for Biological Standards and Control (NIBSC, London, England; code: 16/354). The SARS-CoV-2 alpha variant (B.1.1.7-484K) was isolated from a COVID-19 patient (hCoV-19/Netherlands/UT-RIVM-12844/2021). The SARS-CoV-2 virus culture was performed in Vero E6 cells (Vero C1008, ATCC CRL-1586), which were first grown in Dulbecco's Modified Eagle Medium (DMEM, Gibco) supplemented with 10% fetal bovine serum (FBS) + 1$\times$ penicillin, streptomycin, and glutamine (PSG) at 37°C and 5% $CO_2$. At a confluence of 90 to 95%, the Vero E6 cells were washed twice with PBS, and a virus suspension of SARS-CoV-2 in infection medium (DMEM + 1$\times$ PSG) was added to the cells and incubated at 37°C and 5% $CO_2$ for 2 to 3 days. When a cytopathological effect (CPE) of 90% was observed, the culture supernatant was collected and spun down for 10 min at 1,000 $\times$ $g$ at room temperature (RT) to harvest the virus suspension. Wild-type mumps virus (MuVi/Utrecht.NLD/40.10; genotype G) was cultured on the Vero cells in infection medium containing DMEM + 1$\times$ PSG + 2% FBS, and the virus was harvested when a CPE of >90% was observed. MDCK cells were grown in minimum essential medium (MEM, Gibco) + 10% FBS + 1$\times$ PSG at 37°C and 5% $CO_2$ until a confluence of $\geq$90%. The MDCK cells were washed twice with PBS, and H1N1 virus suspension was added to the cells in MEM + 1$\times$ PSG + 2 $\mu$g/mL L-1-tosylamido-2-phenylethyl chloromethyl ketone (TPCK) treated trypsin. The virus culture was incubated at 37°C and 5% $CO_2$ for 2 to 3 days, and the virus suspension was harvested when a CPE of 90% was observed. Aliquots of each of the virus cultures were snap-frozen and stored at $-80$°C, and viral titers were determined via $TCID_{50}$ assay on Vero E6 cells for SARS-CoV-2 or MDCK cells for H1N1. The SARS-CoV-2 beta variant virus stock was sequenced and did not contain mutations, compared to the reference isolate. The SARS-CoV-2 alpha variant 484K acquired cell culture-induced mutations in amino acids within the open reading frame (ORF) regions: two deletions in

ORF1a:V84- and ORF1a:M85-, and one substitution in ORF1a:L3829F. Due to the mutations, the alpha variant was used only for ELISpot stimulations in which it was not expected to interfere with the experiments.

**Animal handling and sample collection.** The ferrets were anesthetized with ketamine (intramuscular, 5 mg/kg) for weight measurements and swab collection. For drawing blood, infection, and surgical handling, medetomidine (0.1 mg/kg) was added in combination with ketamine and antagonized with antisedan (0.25 mg/kg). Preanalgetic was administered subcutaneously (Carporal, 0.2 mL) prior to the implantation of a transponder for body temperature measurements (Star Oddi, Iceland) in the peritoneal cavity of the ferrets, 20 days before the start of the experiment. Nose and throat swabs were collected in tubes containing viral transport medium (15% sucrose, 2.5 $\mu$g/mL Amphotericin B [Merck], 100 U/mL penicillin, 100 $\mu$g/mL streptomycin, and 250 $\mu$g/mL gentamicin [Sigma]), vortexed, and stored at $-80$°C for later analysis. During the experiment, blood was collected from the cranial vena cava in vacutainer tubes (BD) that were coated with heparin for PBMC isolation or in vacutainer tubes (BD) containing polymer to obtain serum. Euthanasia was performed by exsanguination via heart puncture under anesthesia with ketamine and medetomidine. At necropsy, the nasal turbinates, trachea, and lungs were collected for pathological and virological analysis. The lower portion of the trachea, right nasal turbinates, and three representative lung sections were collected from the right cranial, middle, and caudal lobes in Lysing matrix A tubes (MP Biomedicals, Germany) for virology assays. The trachea and lung samples for virology were stored at $-80$°C until processed for tissue dissociation in infection medium (DMEM + 2% FBS + 1$\times$ PSG). Animals and samples collected up to 9 days after the SARS-CoV-2 infection were handled under BSL-3 conditions. After testing negative for infectious SARS-CoV-2 virus, the samples were handled under BSL-2 conditions. All dissections were performed under BSL-3 conditions. The H1N1-infected samples were handled under BSL-2 conditions.

**Study design and infections.** 12 ferrets were infected intranasally (i.n.) with $10^7$ TCID$_{50}$ of the SARS-CoV-2 beta VOC diluted in phosphate-buffered saline (PBS) on day 0. An inoculum volume of 1 mL was used, with the aim being to reach the lower regions of the respiratory tract. 6 out of 12 ferrets were euthanized at 28 days postinfection (dpi) to describe the postacute phase of SARS-CoV-2 beta VOC infection in ferrets. The remaining six SARS-CoV-2 infected ferrets were infected i.n. with $10^6$ TCID$_{50}$ of H1N1 diluted in PBS. A control group of five ferrets infected only with H1N1 was included. For capacity reasons, the H1N1 infections were performed on two separate days: on 29 dpi (experiment A) and on 30 dpi (experiment B), and they were followed for up to 5 days post-H1N1 infection. The results of experiments A and B are merged, as no cage effects were observed. The mock-infected group of ferrets ($n = 3$) received 1 mL of PBS i.n. only on the days of each different infection (SARS-CoV-2 on day 0, H1N1 on days 29/30), and they were followed up to the end of the experiment (35 dpi). The nose and throat swabs for the virology assays were collected on days 0, 3, 5, and 9 for the SARS-CoV-2 infection and on days 29 to 35 for the H1N1 infection (0, 1, 3, and 5 days after the H1N1 infection). Blood was obtained from the cranial vena cava on days 0, 14, 21, and 29 during the study and by heart puncture on the days of euthanasia (28 and 35 dpi) (Fig. 1A).

**Clinical evaluation.** Clinical examination was performed daily for 9 days post-SARS-CoV-2 infection and for 5 days post-H1N1 infection. Clinical signs were scored by subjective observation using the following scoring guidelines for activity (0, active; 1, active when stimulated; 2, inactive; and 3, lethargic), breathing (0, normal; 1, fast breathing; and 2, heavy/stomach breathing), and the presence (1) or absence (0) of nasal discharge and/or sneezing. Ferrets were euthanized if lethargic or presenting a combined score of 4 for activity and breathing (24). Body temperature was measured by the implanted temperature transponder every 30 min from 5 days prior to each infection. Fever was calculated as temperature deviations from the baseline temperature, measured in degrees Celsius, during the 5 days prior to infection. Variations in body weight were determined relative to the baseline body weight on day 0 of each infection.

**Viral load by RT-qPCR.** Swabs and tissues samples for RT-qPCR were dissociated in lysis buffer containing equine arteritis virus (EAV) as an internal RT-qPCR control. The extraction of total nucleic acid was performed using a MagNA Pure 96 DNA and Viral NA Small Volume Kit in a MagNA Pure 96 system (Roche, Penzberg, Germany) and was eluted in a volume of 50 $\mu$L Roche Tris-HCl elution buffer. A 20 $\mu$L real-time reverse-transcription PCR (RT-qPCR) reaction contained 5 $\mu$L of sample nucleic acid, 7 $\mu$L of 4$\times$ TaqMan Fast Virus Master Mix (Thermo Fisher), 5 $\mu$L of DNase/RNase free water, and 3 $\mu$L of primers and probe mix. SARS-CoV-2 was detected using the E-gene forward (ACAGGTACGTTAATAGTTAATAGCGT) and reverse (ATATTGCAGCAGTACGCACACA) primers and probe (ACACTAGCCATCCTTACTGCGCTTCG). EAV was detected using the EAV forward (CTGTCGCTTGTGCTCAATTTAC) and reverse (AGCGTCCGAAGCATCTC) primers and probe (TGCAGCTTATGTTCCTTGCACTGTGTTC). For H1N1, the primers and probe are designed for the H1N1pmd09 and were detected using the forward (TGGACTTACAATGCCGAACT) and reverse (CAGCG GTTTCCAATTTCCTT) primers and probe (GGACTATCACGGATTCAAATGTGAAGAACT). RT-qPCR was performed for 15 min at 50°C and 2 min at 95°C, followed by 50 cycles of 95°C for 10 s and a primer-specific annealing temperature of 60°C for 30 s, as previously described (24, 45, 46). The subgenomic mRNA of SARS-CoV-2 was detected by in-house subgenomic mRNA E-gene assay, using the E-gene reverse primer and probe and the forward primer, as previously described (47). All experiments were performed on a Light Cycler 480 I (LC480 I, Roche). The cycle threshold (Ct) values were recorded for the genomic and the subgenomic RNA to determine the viral burden and the infectious virus, respectively.

**TCID$_{50}$ determination.** SARS-CoV-2 virus stock, swab material, and tissue samples were 1:10 serial diluted in DMEM medium containing 2% FBS and 1$\times$ PSG and added on Vero E6 cells in 96-well plates. H1N1 virus stock, swab material, and tissue samples were 1:10 serial diluted in MEM medium + 1$\times$ PSG + TPCK treated trypsin (2 $\mu$g/mL) and added on MDCK cells in 96-well plates. The virus stocks were titrated in octuplicate, while the swab material and tissue samples were titrated in sextuplicate. The plates were incubated at 37°C, and CPE was scored after 5 days (H1N1) or 6 days (SARS-CoV-2). The TCID$_{50}$ values were calculated via the Reed-Muench method.

**Histopathology.** Tissues harvested for histological examination (trachea, bronchus, and left lung) were fixed in 10% neutral buffered formalin, embedded in paraffin, sectioned at 4 $\mu$m, and stained with hematoxylin and eosin (H & E) for examination via light microscopy. Histopathology was scored blindly by a veterinary pathologist. The semiquantitative assessment of influenza virus-associated inflammation in the lung was performed on a longitudinal section and a cross-section of the cranial and caudal lobes, as reported earlier (48), with a few modifications. Briefly, for the extent of alveolitis and alveolar damage we scored: 0, 1 (1 to 25%), 2 (25 to 50%), or 3 (>50%). For the severity of alveolitis, bronchiolitis, bronchitis, and tracheitis, we scored: 0, no inflammatory cells; 1, few inflammatory cells; 2, moderate number of inflammatory cells; 3, many inflammatory cells. For the presence of alveolar edema and type II pneumocyte hyperplasia, we scored: 0, 1 (1 to 25%), 2 (25 to 50%), or 3 (>50%). For the presence of alveolar hemorrhage we scored: 0, no; 1, yes. For the extent of peribronchial/perivascular edema, we scored: 0, no; 1, yes. Finally, for the extent of peribronchial, peribronchiolar, and perivascular infiltrates, we scored: 0, none; 1, one to two cells thick; 2, three to ten cells thick; 3, more than ten cells thick. The average scores for the size and the severity of inflammation of the different slides provided the total score per animal.

**Antibody responses.** Antibody responses were measured against the recombinant beta-SARS-CoV-2 B.1.351 spike protein (S) and its receptor binding domain (RBD) (10777-CV and 10735-CV, R&D Systems) via ELISA as previously described (24). Briefly, ferret serum was prediluted 1:50 in PBS + 0.1% Tween 80 solution (dilution buffer), and the ELISA was performed in 2-fold serial dilutions to determine the optimal fitted concentration curve. Next, the plates were incubated with HRP-321 conjugated goat anti-ferret IgG (ab112770, Abcam), diluted 1:5,000 in dilution buffer + 0.5% powdered milk. Color development was obtained via incubation with the KPL Sure Blue TMB Microwell Peroxidase Substrate 1-Component (95059-286, VWR), and the reaction was stopped using 2 M $H_2SO_4$ (80012010.2500, Boom). Antibody concentrations were measured via OD at 450 nm absorbance using an EL808 absorbance reader (Bio-Tek Instruments) and were represented at a 1:100 or 1:200 sera dilution for the spike protein and the RBD, respectively, over the different study time points.

**PBMC isolation.** Peripheral blood mononuclear cells (PBMC) were isolated from 1:1 PBS-diluted blood via density gradient centrifugation (1:1 LymphoPrep 1114547 and Lympholyte-M CL5035, Sanbio). The cells were isolated from the interphase after centrifugation at 800 $\times$ $g$. After two washing steps in RPMI 1640 + 1% FBS, the final cell pellet was resuspended in stimulation medium (RPMI 1640 + 10% FBS + 1$\times$ PSG) at two concentrations: $2.5 \times 10^5$ cells/mL for the ELISpot assays and $3 \times 10^6$/mL for the flow cytometry assay.

**ELISpot.** Pools consisting of 15-mer overlapping (11 amino acids) peptides of entire SARS-CoV-2 proteins from the original SARS-CoV-2 strain (1 $\mu$g/mL, PepMix, JPT), were used for PBMC stimulation. For the spike protein, the peptides were divided over two vials with complementary sequences: 1 to 158 and 159 to 315. The PBMCs were also stimulated with live alpha-SARS-CoV-2 (B.1.1.7) or beta-SARS-CoV-2 (B.1.351) at $2.5 \times 10^4$ TCID$_{50}$ (multiplicity of infection [MOI] of 0.1). The mumps virus ($2.5 \times 10^5$ TCID$_{50}$, MOI of 1) and the HIV peptide pool (1 $\mu$g/mL, PepMix, JPT) were included as negative controls for the live-virus and peptide pool stimulations, respectively. Staphylococcal Enterotoxin B (SEB) was used as a superantigen for lymphocyte activation and served as a positive-control for T cell stimulation. The medium-stimulated condition was included to correct for background activation. The cells were incubated for 20 h at 37°C in 5% $CO_2$ in an incubator in precoated Ferret IFN-$\gamma$-ELISpot plates (3112-4APW-2, Mabtech). The assay was performed following the manufacturer's protocol, with one modification in the incubation time, changing the primary antibody to overnight at 4°C. The plates were air-dried for at least 2 days and were further heat-treated at 65°C for 3 h to ensure the inactivity of SARS-CoV-2 prior to scanning and analysis using an ImmunoSpot S6 CORE (CTL, Cleveland, OH). The background was corrected via the subtraction of medium-stimulated spot counts from every other stimulation.

**IFN-$\gamma$ flow cytometry.** PBMCs at a concentration of $3 \times 10^6$ cells/mL were stimulated with the indicated live virus for 20 h and peptide pools for 6 h at 37°C in 5% $CO_2$ in an incubator. In the last 5 h, Brefeldin A solution (Biolegend) was added to the stimulated PBMCs to prevent the secretion of intracellular cytokines. The stimulated cells were spun down at 500 $\times$ $g$ for 3 min and were resuspended in 150 $\mu$L of washing buffer containing PBS + 0.5% bovine serum albumin + 2 mM EDTA. The cell suspension was transferred into a 96-well plate for flow cytometry. Extracellular staining was performed using APC-conjugated anti-CD4 (60003-MM02-A, clone 02, Sino Biological), eFluor450-conjugated anti-CD8a (17008642, clone OKT8, eBioscience), and FVS780-conjugated cell viability staining for 30 min at 4°C. The cells were washed twice, and intracellular staining was performed using a FoxP3 Staining Kit (eBioscience). Fixation of the cells occurred with the incubation of fixation/permeabilization buffer for 20 min at 4°C, followed by two washing steps with 1$\times$ perm-buffer, prior to incubation for 30 min at RT with the antibodies FITC-conjugated anti-CD3e (MCA1477A647, clone CD3-12, Bio-Rad) and PE-conjugated anti-IFN-$\gamma$ (MCA1783PE, clone CC302, Bio-Rad). Cell acquisition was performed on a BD LSRFortessa Cell Analyzer (BD Biosciences) and analyzed using FlowJo V10.6.2 (BD Biosciences).

**Statistical analysis.** All of the data were analyzed using the GraphPad Prism 9.1.0 software package (GraphPad Software, Inc.). Normality tests were employed to define the statistical tests for the different data. Numerical values representing absolute or relative numbers are presented by means and standard deviations or by medians and interquartile ranges, according to the Gaussian distribution. Comparisons between two groups were performed using the Mann-Whitney test or Student's $t$ test. Comparisons between multiple groups were performed using a one-way analysis of variance (ANOVA) or the Kruskal-Wallis test. Correction for multiple comparisons was done using Tukey's multiple-comparison test. The total area under the curve was calculated for the body temperature measures across the groups. Fisher's exact test was employed to analyze the statistical significance of the categorical data of the clinical signs. The threshold for statistical significance was set at $P < 0.05$.

## SUPPLEMENTAL MATERIAL

Supplemental material is available online only.

**SUPPLEMENTAL FILE 1**, PDF file, 0.1 MB.

## ACKNOWLEDGMENTS

We thank Helena Pinheiro Guimarães for her help during the animal experiments, Maarten Emmelot and Gabriel Goderski for providing the mumps and SARS-CoV-2 viruses, and Elena Pinelli Ortiz and Willem Luytjes for their critical review of the manuscript.

This study was funded by the Dutch Ministry of Health, Welfare, and Sports (VWS).

We declare no competing interests.

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
