## [Reviewer comments · Microbiology Spectrum]

Microbiology Spectrum

Influenza infection in ferrets with SARS-CoV-2 infection history

Caroline de Melo, Florence Peters, Harry van Dijken, Stefanie Lenz, Koen van de Ven, Lisa Wijsman, Angela Gomersbach, Tanja Schouten, Puck van Kasteren, Judith van den Brand, and Jorgen de Jonge

Corresponding Author(s): Jorgen de Jonge, National Institute for Public Health and the Environment

Review Timeline:

Submission Date:	May 6, 2022
Editorial Decision:	June 20, 2022
Revision Received:	September 8, 2022
Accepted:	September 23, 2022

Editor: Daniela Rajao

Reviewer(s): The reviewers have opted to remain anonymous.

Transaction Report:

DOI: <https://doi.org/10.1128/spectrum.01386-22>

June 20, 2022

Dr. Jorgen de Jonge
National Institute for Public Health and the Environment
Antonie van Leeuwenhoeklaan 9
Bilthoven
Netherlands

Re: Spectrum01386-22 (Influenza infection in ferrets with SARS-CoV-2 infection history)

Dear Dr. Jorgen de Jonge:

Link Not Available

Sincerely,

Daniela Rajao

Journals Department
Reviewer comments:

Reviewer #1 (Comments for the Author):

This revised manuscript by de Melo et al describes the clinical, histopathological, and immunological effects of influenza A virus (IAV) infection following recovery from infection with the beta variant of concern (VOC) of SARS-CoV-2 using a ferret model. Following beta VOC infection, ferrets showed minimal disease development and lung pathology and low-grade viral replication. Upon subsequent IAV infection, previous SARS-CoV-2 infection led to mild, nonsignificant increases in severity of clinical disease and lung pathology relative to IAV-infected, SARS-CoV-2-naïve ferrets. This is an important and timely topic, as IAV infection following SARS-CoV-2 infection will undoubtedly be commonplace in the coming years. While this manuscript is very well written with well-designed experiments, the authors may consider the following as opportunities for improvement.

1. Previous reviews of the manuscript criticized the use to the beta VOC, as it produced low-grade infection and minimal clinical disease in ferrets; the authors justified its use as it was a circulating VOC at the time of the study. However, a large proportion of humans infected with SARS-CoV-2 demonstrate minimal clinical disease or are asymptomatic, and this population has the potential of being infected with IAV at a later date just as much as patients who recover from moderate to severe COVID-19. Furthermore, many people who had asymptomatic COVID have reported long term clinical symptoms following infection, and subsequent IAV infection may affect the severity or resolution of these symptoms. Discussion of this concept should be added to the manuscript.

2. Use of the term "symptoms" when describing clinical disease in ferrets is not technically correct, as the term refers to a disease state that is apparent to the patient. Because animals cannot verbalize their specific experience, the term "clinical signs" should be used instead.

3. Please clarify whether ferrets were socially housed during the experiments and if so, whether there were any cage-wide effects within experimental groups.

4. Figure 4G: Please change the data presentation in this figure from a single data point per group to column graphs.

5. Figure 5C: The photomicrographs are too small and of too poor quality to distinguish the different pathological features. Please enlarge the images within the figure and include a magnified inset to better indicate the areas of type II pneumocyte hyperplasia.

Reviewer #2 (Comments for the Author):

The manuscript explores the characteristics of a sequential inoculation of SARS-CoV-2 and Influenza using the widely described ferret model. Since most non-therapeutic interventions are easing worldwide, it is imperative to understand the interactions that different respiratory viruses can have with SARS-CoV-2. The manuscript is well-written, with a clear rationale behind the experiments. Unfortunately, one of the main factors required to properly investigate the impact of SARS-CoV-2 over a subsequential influenza infection, an active SARS-CoV-2 infection, could not be achieved, affecting the results, and narrowing the conclusions that can be obtained from the experiments performed. Regardless of this, the information provided is of general interest to the research community.

Major comments

A significant limitation of the study is the poor replication of beta observed in ferrets (as previous reviewers mentioned). Were any standard curves run through RT-qPCR? Only the Ct values make it challenging to put in context the level of replication determined by PCR observed in ferrets?. A standard curve that will allow to transform of those Ct values into TCID50 equivalent would facilitate the interpretation of the results. Furthermore, little virus detected by TCID50 makes me wonder whether ferrets got actively infected. The tissue data supports my doubts about active replication of SARS2 in the ferrets and suggests that any detection beyond Ct of 35 is negligible since those tissues were collected on day 28 when the infection was most likely cleared. The authors addressed the issues with the productive infection; however, this is a critical step in the manuscript that unfortunately doesn't allow the generation of proper conclusions since it is difficult to establish that ferrets were infected and SARS2 productively replicated as it is shown currently. Despite the lack of high replication, the authors successfully demonstrate cellular and humoral responses against SARS. Therefore, the results suggest an exposure rather than an active infection/replication of SARS, which would require modifications in the text that reflect this. Infection history based on the results presented is not the most accurate term.

The lack of representative histology figures that show the finding described must be addressed. Panels showing at least mock-infected animals plus infected animals should be included. I can understand if the authors focus just on the major findings for the pictures, but at least those should be included in the figures.

Showing the gross pathology upon influenza infection will facilitate to the readers the conclusions obtained (line 371-375)

Minor comments

Figure 2B: Which units were used to evaluate body temperature?

Please discuss the lack of response to nsp12 (fig 3)

Is the data presented in figure 4G independent of the magnitude of the clinical sign evaluated? Was nasal discharge and sneezing considered? Any differences observed?

Why were male ferrets used when previous reports showed successful replication in female ferrets?

Line 490: There is an extra dot.

Please discuss more in detail the study's limitations regarding the poor replication of the beta VOC. What could be done to overcome this limitation? How is the replication of more recent VOC in ferrets?

The text often mentions the long-covid syndrome or post-acute COVID-19; however, such a condition was not replicated in the ferret model. I suggest modifying the manuscript to decrease the references to this phenomenon.

Modify IAV for the most current nomenclature: FLUAV

Reviewer #4 (Comments for the Author):

Vilas Boas de Melo et al., sought to determine the effect a prior SARS-CoV-2 infection would have on disease severity when animals are subsequently infected with H1N1 influenza. Previous studies have examined co-infections with SARS-CoV-2 and influenza, but whether previous SARS-CoV-2 infection would result in different disease outcomes upon influenza infection was yet to be determined. In this study, ferrets were first infected with SARS-CoV-2 and then infected 28 days later with H1N1 influenza. The initial SARS-CoV-2 infection did not result in disease and the virus replicated to low levels in the nose and throat of the animals. Despite this relatively mild infection, the authors demonstrate SARS-CoV-2 infection induced both humoral and cellular immunity. Upon H1N1 challenge, prior SARS-CoV-2 infection did not result in any significant changes in influenza replication; however, there was an increase in frequency of clinical signs as well as a trend towards more severe bronchitis. These findings suggest prior SARS-CoV-2 infection may enhance disease severity upon influenza infection.

Major Comments:

1. At several places in the manuscript (Abstract lines 34-45, Importance, lines 47-48, Introduction lines 101-102), the authors indicate that their findings provide evidence supporting the use of influenza vaccination to mitigate the effects of influenza infection in patients with long-COVID. The manuscript does not examine this question as vaccination is not part of the manuscript and there is no definitive data in the manuscript that supports this claim.
2. The authors make several references to long-COVID or post-acute COVID and that the ferret is a model of this condition. The clinical definition of long-COVID is vague, but most sources agree that long-COVID consists of signs or symptoms that persist beyond 1-month post-acute infection. The findings in the manuscript do not indicate that ferrets are experiencing prolonged signs and there is no indication of residual SARS-CoV-2 induced disease. Thus, it is not accurate to indicate that ferrets are modeling this scenario. The authors are instead modeling influenza infection after recovery from a mild-SARS-CoV-2 infection.
3. One of the main findings of the paper was a difference in the frequency of clinical scores between groups. Therefore, it would be beneficial to describe how this scoring was determined in greater detail. Breathing scores were described as "normal" or "fast", but how was this measured? For how long and how frequently was breathing or activity observed? Were the breaths/minute counted or was this a subjective observation? Clinical signs were shown on days 4 and 5 in Figure 4g- were there any clinical symptoms shown on days 1, 2, or 3?

Minor Comments:

1. The manuscript should be revised for grammar and word usage. For example, in line 60. The correct term for removing NPI's is "relaxed" or "ceased". Line 607, please replace "injected" i.n. with "inoculated or instilled" i.n.
2. There are discrepancies in the timeline of the influenza infection and when samples were taken: Figure 1 shows day 2, 3, and 5. Methods indicate 1, 3, and 5 days post infection (line 184). Results indicate 2, 4, and 5 days post infection (line 368).
3. Lines 391-405. It would strengthen this part of the manuscript to clearly indicate that animals euthanized on day 28 after SARS-CoV-2 infection had no evidence of residual bronchiolitis at the time of H1N1 infection.
4. Lines 480-483. It is unclear what is meant by "SARS-CoV-2-specific cellular and humoral responses were associated with protective immunity in human COVID-19, which may have also contributed to the controlled SARS-CoV-2 infection in ferrets". This sentence appears to indicate that somehow immunity in humans confers immunity in ferrets. Please revise to clarify this sentence. As well, if the authors are claiming that cellular and humoral immunity controlled the SARS-CoV-2 infection in ferrets, this is likely true; however, it is also important to highlight that the ferret is a semi-permissive model of SARS-CoV-2 and this likely also contributed to the mild infection.
5. Lines 484, please explain the term "trained immunity" in more detail. This is not a commonly used term, and it would improve interpretation. In addition, please also describe what is meant by "heterologous" infection, is this different influenza or SARS-CoV-2 strains, or is this influenza vs SARS-CoV-2.
6. In Figure 1a and b, not all 12 animals were positive for vRNA in the nose and throat swabs. Please indicate the proportion of

animals that were positive at the time points when no vRNA was detected.

7. Figure 3C. Are there significant differences in the response to beta-SARS-CoV-2 compared to the other responses? If there are differences this should be denoted. If not, this should be explained in the text.

Staff Comments:

Preparing Revision Guidelines

Please return the manuscript within 60 days; if you cannot complete the modification within this time period, please contact me. If you do not wish to modify the manuscript and prefer to submit it to another journal, please notify me of your decision immediately so that the manuscript may be formally withdrawn from consideration by Microbiology Spectrum.

Response to reviewers:

Thank you for considering our manuscript entitled "Influenza infection in ferrets with SARS-CoV-2 infection history" (Spectrum 01386-22) for publication in Microbiology Spectrum. We appreciate the critical analyses of the editor and the reviewers. We believe that all concerns have been appropriately addressed and explained in the revised version of the manuscript and in this letter. Please find our responses to the comments in red below.

Reviewer #1 (Comments for the Author):

This revised manuscript by de Melo et al describes the clinical, histopathological, and immunological effects of influenza A virus (IAV) infection following recovery from infection with the beta variant of concern (VOC) of SARS-CoV-2 using a ferret model. Following beta VOC infection, ferrets showed minimal disease development and lung pathology and low-grade viral replication. Upon subsequent IAV infection, previous SARS-CoV-2 infection led to mild, nonsignificant increases in severity of clinical disease and lung pathology relative to IAV-infected, SARS-CoV-2-naïve ferrets. This is an important and timely topic, as IAV infection following SARS-CoV-2 infection will undoubtedly be commonplace in the coming years. While this manuscript is very well written with well-designed experiments, the authors may consider the following as opportunities for improvement.

1. Previous reviews of the manuscript criticized the use to the beta VOC, as it produced low-grade infection and minimal clinical disease in ferrets; the authors justified its use as it was a circulating VOC at the time of the study. However, a large proportion of humans infected with SARS-CoV-2 demonstrate minimal clinical disease or are asymptomatic, and this population has the potential of being infected with IAV at a later date just as much as patients who recover from moderate to severe COVID-19. Furthermore, many people who had asymptomatic COVID have reported long term clinical symptoms following infection, and subsequent IAV infection may affect the severity or resolution of these symptoms. Discussion of this concept should be added to the manuscript.

We thank the reviewer for this insight. We agree that this is a valuable different perspective and have added the following to the discussion: "The low level of virus replication and absence of disease in our beta VOC model may on the other hand represent a large part of the human population that undergo mild or asymptomatic SARS-CoV-2 infections and that are also subject to subsequent influenza infections". (Line 476-479)

2. Use of the term "symptoms" when describing clinical disease in ferrets is not technically correct, as the term refers to a disease state that is apparent to the patient. Because animals cannot verbalize their specific experience, the term "clinical signs" should be used instead.

The reviewer is correct. We replaced "symptoms" by "clinical signs" where referring to ferrets.

3. Please clarify whether ferrets were socially housed during the experiments and if so, whether there were any cage-wide effects within experimental groups.

The ferrets were housed according to the groups described in the manuscript (Ln 117-120), except during infection period, when the groups were further divided into 3 ferrets/isolator (now included in Ln-119-120). No cage effects were observed (included in line 181)

4. Figure 4G: Please change the data presentation in this figure from a single data point per group to column graphs.

For objective representation of the data into a single graph, we changed the Y-axis to absolute number of ferrets, to represent the individual observations as suggested by the reviewer. We also adjusted the data to columns to improve visualization.

5. Figure 5C: The photomicrographs are too small and of too poor quality to distinguish the different pathological features. Please enlarge the images within the figure and include a magnified inset to better indicate the areas of type II pneumocyte hyperplasia.

We agree with the reviewer and adjusted the figure 5C accordingly

Reviewer #2 (Comments for the Author):

The manuscript explores the characteristics of a sequential inoculation of SARS-CoV-2 and Influenza using the widely described ferret model. Since most non-therapeutic interventions are easing worldwide, it is imperative to understand the interactions that different respiratory viruses can have with SARS-CoV-2. The manuscript is well-written, with a clear rationale behind the experiments. Unfortunately, one of the main factors required to properly investigate the impact of SARS-CoV-2 over a subsequential influenza infection, an active SARS-CoV-2 infection, could not be achieved, affecting the results, and narrowing the conclusions that can be obtained from the experiments performed. Regardless of this, the information provided is of general interest to the research community.

Major comments

A significant limitation of the study is the poor replication of beta observed in ferrets (as previous reviewers mentioned). Were any standard curves run through RT-qPCR? Only the Ct values make it challenging to put in context the level of replication determined by PCR observed in ferrets?. A standard curve that will allow to transform of those Ct values into TCID50 equivalent would facilitate the interpretation of the results.

We thank the reviewer for the critical look and comments, which allowed us to sharpen the text. With regard to RT-qPCR Ct values, these cannot be extrapolated to TCID50, as RT-qPCR detects viral RNA that is present in (apoptotic)cells, noninfectious and infectious particles. The ratio's between these depend on the infection phase. For example, viral RNA from cells is still detected long after infectious virus is gone. Thus these two assays do not correlate in the same way at all times during an infection. The RT-qPCR data should be regarded as a semi-quantitative measure to determine when virus replication peaked and decreased. We performed a TCID50 assay to assess the infectious viral load.

Furthermore, little virus detected by TCID50 makes me wonder whether ferrets got actively infected. The tissue data supports my doubts about active replication of SARS2 in the ferrets and suggests that any detection beyond Ct of 35 is negligible since those tissues were collected on day 28 when the infection was most likely cleared. The authors addressed the issues with the productive infection; however, this is a critical step in the manuscript that unfortunately doesn't allow the generation of proper conclusions since it is difficult to establish that ferrets were infected and SARS2 productively replicated as it is shown currently. Despite the lack of high replication, the authors successfully demonstrate cellular and humoral responses against SARS. Therefore, the results suggest an exposure rather than an active infection/replication of SARS, which would require modifications in the text that reflect this. Infection history based on the results presented is not the most accurate term.

We do not agree with the reviewer that there was probably no virus replication. The following findings are to our opinion proof of a productive infection, although indeed a very low level of replication likely occurred:

- An increase in TCID50 and subgenomic viral RNA (measured by qPCR) in the nose from day 3 to day 5.
- Presence of replication-competent virus at day 5 cannot be input virus as this would have degraded and/or have been cleared by the innate immune system by that time.
- Ct values below 30 are detected consistently in the throat until Day 9 post infection. A Ct value of 30 is generally accepted as clearly positive.
- Detection of subgenomic RNA, which only occurs when cells are infected by SARS-CoV-2 as this type of RNA is produced in the cell to transcribe viral proteins and is not included in the virus particle.
- We previously showed that exposure to SARS-CoV-2 without detectable replication did not result in an immune response (doi: 10.3389/fimmu.2021.750229). Thus, the observation of a clear humoral and cellular immune response in the current study is indicative for active viral replication.

We added the following statement to lines 378-381: "Altogether, the viral replication and immune data indicate that a productive, but low-level, replication was established in the majority of the ferrets as only exposure to virus does not lead to an immune response [24]."

And we changed the discussion to further clarify this point: "Using a high infectious dose (10^7 TCID50/mL) in male ferrets, we found that the beta VOC replicated only to low levels as measured by TCID50. Productive infection was further confirmed by the detection of subgenomic viral RNA and clear development of a humoral and cellular immune response, which does not occur when only exposed to virus [24]." Line (490-495)

Furthermore, it is expected that the tissues are negative at D28 as then virus has normally disappeared. We clarified that better in the text now (lines 341-345)

The lack of representative histology figures that show the finding described must be addressed. Panels showing at least mock-infected animals plus infected animals should be included. I can understand if the authors focus just on the major findings for the pictures, but at least those should be included in the figures.

We are not sure what the reviewer means by lack of histopathology, as in figure 5 we present mock vs infected animals. In any case these figures have been further improved by adding a magnification to show some details. We did not include histopathology figures of animals 4 weeks after SARS-CoV-2 infection as they principally would illustrate normal tissue.

Showing the gross pathology upon influenza infection will facilitate to the readers the conclusions obtained (line 371-375)

We aim to highlight differences between the influenza-infected ferrets with or without previous SARS-CoV-2 exposure – and to that effect, no differences were observed. H1N1 influenza infection and gross-pathology has been described quite extensively before and since we did not find any differences, showing gross pathology seems redundant.

Minor comments

Figure 2B: Which units were used to evaluate body temperature?

We added (°C) to line 199.

Please discuss the lack of response to nsp12 (fig 3)

This is now discussed in lines 508-509; It is consistent with findings in humans that this protein does not induce a high frequency of T-cells.

Is the data presented in figure 4G independent of the magnitude of the clinical sign evaluated? Was nasal discharge and sneezing considered? Any differences observed?

Nasal discharge and sneezing were also considered (See Methods, 192-194), but only decreased behavioral activity and increased severity of breathing were observed. We better clarified in the figure legend of Fig4G the magnitude and frequency: 'Indicated are the number of ferrets that had decreased activity (score 1) and labored breathing (score 1) according to the scoring system described in the materials and methods section.' (Lines 772-774).

Why were male ferrets used when previous reports showed successful replication in female ferrets?

At the time, only one report had shown successful replication in female ferrets with the beta VOC, and there were no reports of sex being a limiting factor for an active infection. In addition, we had successfully performed a SARS-CoV-2 study in male ferrets before.

Line 490: There is an extra dot.

Thank you for the observation, this was corrected.

Please discuss more in detail the study's limitations regarding the poor replication of the beta VOC. What could be done to overcome this limitation? How is the replication of more recent VOC in ferrets?

We adjusted the discussion regarding these point raised in lines 472-479, 492-495 and from line 495 and onward different VOC are discussed and it is mentioned that e.g. the alpha VOC does replicate well. This strain would obviously be an alternative.

The text often mentions the long-covid syndrome or post-acute COVID-19; however, such a condition was not replicated in the ferret model. I suggest modifying the manuscript to decrease the references to this phenomenon.

We indeed do not show that the ferret is a model for long-COVID. For example we state that: “We could not induce clear long-term COVID-19 effects as SARS-CoV-2 infection in ferrets was mild (Line 44-45)”. Our intention is to hypothesize and discuss this option as it was a reason to select the ferret model. Therefore we refer to one study in line (93-95) in the introduction: “These observations could reflect, in part, the long-term disease (post-acute COVID-19) characterized by prolonged respiratory complaints and fatigue.” This study suggested that the ferrets **could** be a model. We added one sentence to the discussion to make clear that we do not claim the ferret as a model for long-COVID: Thus, using the beta VOC, we could not induce any histopathological effects that would be indicative for long-COVID (Line 472-473).

Modify IAV for the most current nomenclature: FLUAV

We changed IAV into FLUAV

Reviewer #4 (Comments for the Author):

Vilas Boas de Melo et al., sought to determine the effect a prior SARS-CoV-2 infection would have on disease severity when animals are subsequently infected with H1N1 influenza. Previous studies have examined co-infections with SARS-CoV-2 and influenza, but whether previous SARS-CoV-2 infection would result in different disease outcomes upon influenza infection was yet to be determined. In this study, ferrets were first infected with SARS-CoV-2 and then infected 28 days later with H1N1 influenza. The initial SARS-CoV-2 infection did not result in disease and the virus replicated to low levels in the nose and throat of the animals. Despite this relatively mild infection, the authors demonstrate SARS-CoV-2 infection induced both humoral and cellular immunity. Upon H1N1 challenge, prior SARS-CoV-2 infection did not result in any significant changes in influenza replication; however, there was an increase in frequency of clinical signs as well as a trend towards more severe bronchitis. These findings suggest prior SARS-CoV-2 infection may enhance disease severity upon influenza infection.

Major Comments:

1. At several places in the manuscript (Abstract lines 34-45, Importance, lines 47-48, Introduction lines 101-102), the authors indicate that their findings provide evidence supporting the use of influenza vaccination to mitigate the effects of influenza infection in patients with long-COVID. The

manuscript does not examine this question as vaccination is not part of the manuscript and there is no definitive data in the manuscript that supports this claim.

We thank the reviewer for critically reviewing the manuscript.

It is not our intention to claim that we show that influenza vaccination protects against enhanced effects in long-COVID patients. However, as influenza vaccination prevents virus infection and one may expect that the risk of enhanced disease as a result of a viral infection is also reduced. Therefore we state that it is advisable to consider vaccination as a mitigation strategy. We did remove line 101-102 as indeed this sentence appears to claim that vaccination is part of the study as indicated by the reviewer.

2. The authors make several references to long-COVID or post-acute COVID and that the ferret is a model of this condition. The clinical definition of long-COVID is vague, but most sources agree that long-COVID consists of signs or symptoms that persist beyond 1-month post-acute infection. The findings in the manuscript do not indicate that ferrets are experiencing prolonged signs and there is no indication of residual SARS-CoV-2 induced disease. Thus, it is not accurate to indicate that ferrets are modeling this scenario. The authors are instead modeling influenza infection after recovery from a mild-SARS-CoV-2 infection.

We indeed do not show that the ferret is a model for long-COVID. For example we state that: “We could not induce clear long-term COVID-19 effects as SARS-CoV-2 infection in ferrets was mild (Line 44-45)”. Our intention is to hypothesize and discuss this option as it was a reason to select the ferret model. Therefore we refer to one study in line (93-95) in the introduction: “These observations could reflect, in part, the long-term disease (post-acute COVID-19) characterized by prolonged respiratory complaints and fatigue.” This study suggested that the ferrets **could** be a model. We added one sentence to the discussion to make clear that we do not claim the ferret as a model for long-COVID: Thus, using the beta VOC, we could not induce any histopathological effects that would be indicative for long-COVID (Line 472-473).

3. One of the main findings of the paper was a difference in the frequency of clinical scores between groups. Therefore, it would be beneficial to describe how this scoring was determined in greater detail. Breathing scores were described as "normal" or "fast", but how was this measured? For how long and how frequently was breathing or activity observed? Were the breaths/minute counted or was this a subjective observation? Clinical signs were shown on days 4 and 5 in Figure 4g- were there any clinical symptoms shown on days 1, 2, or 3?

We agree that this was not clearly described for the clinical observation and was focused on the endpoints. We now changed this (Line: 192-198).

A sentence was included in Lines 400-401 to describe that the observed clinical signs were only detected from day 4 following H1N1 infection.

And explanatory text was added to the legend “Recorded clinical signs of influenza disease. Indicated are the number of ferrets that had decreased activity (score 1) and labored breathing (score 1) according to the scoring system described in the materials and methods section.” (Line 772-774)

Minor Comments:

1. The manuscript should be revised for grammar and word usage. For example, in line 60. The correct term for removing NPI's is "relaxed" or "ceased". Line 607, please replace "injected" i.n. with "inoculated or instilled" i.n.

We replaced release by relaxation and injected by administered.

2. There are discrepancies in the timeline of the influenza infection and when samples were taken: Figure 1 shows day 2, 3, and 5. Methods indicate 1, 3, and 5 days post infection (line 184). Results indicate 2, 4, and 5 days post infection (line 368).

Thank you for the notification. This has been corrected to days 1, 3 and 5 in figures 1 and 4 and in-text (Lines 384-387).

3. Lines 391-405. It would strengthen this part of the manuscript to clearly indicate that animals euthanized on day 28 after SARS-CoV-2 infection had no evidence of residual bronchiolitis at the time of H1N1 infection.

We adjusted accordingly (lines 411-413); "Although we did not observe any strong pathological effects in the lungs one day (D28) prior to influenza infection in ferrets with a SARS-CoV-2 infection history (Fig 2d,e), a previous SARS-CoV-2 infection did appear to marginally worsen influenza disease signs."

4. Lines 480-483. It is unclear what is meant by "SARS-CoV-2-specific cellular and humoral responses were associated with protective immunity in human COVID-19, which may have also contributed to the controlled SARS-CoV-2 infection in ferrets". This sentence appears to indicate that somehow immunity in humans confers immunity in ferrets. Please revise to clarify this sentence.

As well, if the authors are claiming that cellular and humoral immunity controlled the SARS-CoV-2 infection in ferrets, this is likely true; however, it is also important to highlight that the ferret is a semi-permissive model of SARS-CoV-2 and this likely also contributed to the mild infection.

Thank you for this remark. We changed the conclusion and scope of this paragraph as it is more relevant to state that the observed cellular responses reflect those observed in humans at the protein level. This is an indication that the ferret model may be suitable for studying cellular responses (lines 508-511).

5. Lines 484, please explain the term "trained immunity" in more detail. This is not a commonly used term, and it would improve interpretation. In addition, please also describe what is meant by "heterologous" infection, is this different influenza or SARS-CoV-2 strains, or is this influenza vs SARS-CoV-2.

It now reads: The innate system has its own memory-like capability, called trained immunity. An encounter with a pathogen keeps the innate immune system in a more alert state for a period of time. Trained immunity elicited by a priming infection is therefore also considered to play a role in the protection against heterologous subsequent infections by other pathogens. (Lines 514-515)

6. In Figure 1a and b, not all 12 animals were positive for vRNA in the nose and throat swabs. Please indicate the proportion of animals that were positive at the time points when no vRNA was detected.

We adjusted this in lines 327-336.

7. Figure 3C. Are there significant differences in the response to beta-SARS-CoV-2 compared to the other responses? If there are differences this should be denoted. If not, this should be explained in the text.

We did not do a statistical test as these responses cannot be compared as they are elicited with different stimuli (live virus vs peptide pools). Live virus actively infects cells, through which MHC presentation is established and triggers additional innate pathways that could enhance the specific response.

September 23, 2022

Dr. Jorgen de Jonge
National Institute for Public Health and the Environment
Antonie van Leeuwenhoeklaan 9
Bilthoven
Netherlands

Re: Spectrum01386-22R1 (Influenza infection in ferrets with SARS-CoV-2 infection history)

Dear Dr. Jorgen de Jonge:

Your manuscript has been accepted, and I am forwarding it to the ASM Journals Department for publication. You will be notified when your proofs are ready to be viewed.

Sincerely,

Daniela Rajao
Editor, Microbiology Spectrum
